# Mechanical Properties and Microstructure of Austenite—Ferrite Duplex Stainless Steel Hybrid (Laser + GMAW) and SAW Welded Joint

**DOI:** 10.3390/ma16072909

**Published:** 2023-04-06

**Authors:** Ryszard Krawczyk, Jacek Słania, Grzegorz Golański, Tomasz Pfeifer

**Affiliations:** 1Faculty of Mechanical Engineering and Computer Science, Czestochowa University of Technology, Armii Krajowej 21, 42-201 Czestochowa, Poland; 2Łukasiewicz Research Network Upper Silesian Institute of Technology, K. Miarki 12-14, 44-100 Gliwice, Poland; 3Department of Material Engineering, Czestochowa University of Technology, Armii Krajowej 19, 42-201 Czestochowa, Poland

**Keywords:** duplex steel, HLAW + SAW welded joint, microstructure, mechanical properties

## Abstract

The purpose of the research was to develop a technology for producing thick-walled duplex steel welded joints. The material used in the research was X2CrNiMoN22 duplex steel in the form of a 15 mm thick plate. The welded joint was produced by the modern, high-performance Hybrid Laser Arc Welding (HLAW) method. The HLAW method involves welding a joint using a laser, the Gas Metal Arc Welding (GMAW) method and the Submerged Arc Welding (SAW) method. The HLAW method was used to make the root pass of the double butt welded joint, while the filler passes were made by the SAW method. The obtained welded joint was subjected to non-destructive and destructive testing. The non-destructive and macroscopic tests allowed the joint to be classified to the quality level B. Microscopic examinations revealed the presence of ferritic–austenitic microstructure in the base material and the weld, with different ferrite content in specific joint areas. The analysed joint had high strength properties (tensile strength (TS) ~ 790 ± 7 MPa) and high ductility of weld metal (~160 ± 4 J) heat-affected zone (~216 ± 26 J), and plasticity (bending angle of 180° with no macrocracks). At the same time, hardness on the cross-section of the welded joint did not exceed 280 HV10.

## 1. Introduction

Duplex steels are corrosion-resistant steels with approx. 50% ferrite and 50% austenite. They have favourable mechanical properties (strength and toughness) in the service temperature range from −50 to 280 °C, good general corrosion and pitting resistance, stress corrosion cracking resistance and adequate weldability and formability. The performance of the duplex steel depends not only on the content of alloying elements, but mainly the content of ferrite and austenite in the microstructure. Compared to conventional austenitic steels, duplex steels have also a more attractive price as a construction material due to lower content of expensive nickel. Due to their valuable performance, duplex steels are used in various branches of modern industry, especially in the chemical, food, petrochemical and construction industries [1,2].

The welding of duplex steel structures and components entails the need to take into account not only the impact of chemical composition of the filler metal and base material, but also the thermal welding conditions on the composition of structure, the volume fraction of the phases—ferrite and austenite in the heat-affected zone (HAZ) and the weld. For welded joints, the important parameter that determines the welded joint structure is the cooling rate associated in a significant way with the welding conditions. Therefore, when welding duplex steel, particular attention should be paid to the heat input also in direct correlation with the shielding gas composition, the preparation of the joint for welding and the method for depositing beads [3,4,5,6].

The increase in the cooling rate of a duplex steel welded joint contributes to inhibition of the ferrite-to-austenite conversion, resulting in a higher volume fraction of ferrite. This metallurgical regularity is found in the weld and the HAZ of the welded joint where a coarse-grained microstructure with high ferrite content can be formed as a result of incorrectly selected parameters and too fast cooling after welding. This structure has low plasticity and low corrosion resistance. It is assumed that duplex steels demonstrate their beneficial properties when ferrite content is not lower than 25–30% [5,7].

When welding thick-walled duplex or superduplex steel components, a significant problem is the precipitation of, among others, harmful intermetallic phases, e.g., chi—or sigma—phases, within the weld volume or HAZ due to multi-pass welding [8,9,10]. The appearance of these phases contributes not only to the occurrence of brittleness, but also has a negative influence on the welded joint corrosion resistance. One of the methods to limit the likelihood of these harmful phases being precipitated is to minimise the amount of energy introduced during welding (arc energy) and use the appropriate interpass temperature [8,10]. The selection of the appropriate welding heat input is an important element for making a correct welded joint in duplex stainless steel. Welding with low arc energy provides rapid cooling, which, however, can result in the formation of a structure with too much ferrite in the weld and the HAZ. The increase in the proportion of ferrite in these areas can lead to deterioration of the plastic properties and decrease in the corrosion resistance of the welded joint. On the other hand, welding made with too high heat input can contribute to the precipitation and growth of unfavourable hard and brittle phases due to the extension of the HAZ residence time at above 280 °C [7,8,10,11].

In production conditions, there is a need for high-performance welding processes, especially for thick plates. This involves the introduction of a significant amount of heat into the joint. This heat may adversely affect the phase structure of the welded joint. The purpose of the non-destructive and destructive tests carried out in the work and the analysis of the results obtained was to confirm the usefulness of the high-performance combined process (hybrid welding and SAW) for welding thick-walled X2CrNiMoN22-5-3 plates.

## 2. Materials and Research Methodology

The analysis was performed on a thick-walled duplex steel welded joint. The prepared joint was a butt joint of X2CrNiMoN22-5-3 steel plates with a thickness of t = 15 mm. The chemical composition of the test base material determined with the Bruker XP spectrometer is presented in Table 1.

The test butt joint was prepared with a double-side bevel (2Y), a 5 mm high threshold and a groove angle of 70°. The test joint was made under automated conditions using the hybrid welding process, i.e., HLAW and classic SAW welding (Figure 1).

The following equipment was used for the hybrid welding:KUKA KR30HA welding robot fitted with Trumpf TruDisk 12,002 laser with Trumpf D70 head for laser welding;EWM welding power supply with a push-pull feeder for MAG welding.

For submerged arc welding, the LAH 1001 DC power supply fitted with the A2 Multitrack welding tractor with A2/A6 PEK digital process controller was used. The filler material was OK Autrod 2209 welding electrode, whose chemical composition and mechanical properties based on the manufacturer’s certificate are presented in Table 2.

For the MAG method of the hybrid welding process, the ESAB OK Autrod 2209 welding electrode and M12 shielding gas (ArCo_2_ with the active gas content of 2.5%) were used, while for the submerged arc welding, the ESAB OK Autrod 2209 welding electrode and ESAB OK FLUX 10.93 were used. The schematic diagram of the hybrid + SAW welding process used is presented in Figure 1.

The first bead was deposited in the central zone of the joint by the hybrid (laser + GMAW) method in the L-A system, while the second and third ones were made as single-pass filler welds by the SAW method. The heat input during welding by the hybrid (laser + GMAW) and SAW methods was 0.83 and 1.54 kJ/mm, respectively. Before welding the drying heating was applied at 300 °C for two hours and during welding the recommended interpass temperature of 110 °C was controlled.

Non-destructive testing, i.e., visual tests (VT), penetrant tests (PT) and radiographic tests (RT) were performed to evaluate the quality of the welded joint. The VT, PT and RT were made on both sides of the weld over the entire length of the joint in accordance with the requirements of the relevant standards, e.g., [12,13]. For evaluation of the experimental joint, the quality level B was adopted. The evaluation of the mechanical properties of the test joint included: macroscopic examinations, static tensile test, side bend test, impact test and hardness measurement.

The tensile and bend tests were carried out with the MTS 800 testing machine. The minimum value of 660 N/mm^2^ was adopted for evaluation of the tensile strength (TS) and the bending angle of 180° was used for the bend test evaluations. The diameter of the bending mandrel was 50 mm. The impact energy test on standard Charpy V-notch samples taken from two areas, i.e., in the weld and the heat-affected zone (HAZ), was carried out with the WOLPERT W-15 impact testing machine. The impact energy was evaluated taken transversely from each of the zones of the welded joint, i.e., weld (VWT 0/2) and HAZ (VHT 0/2), based on their average values according to the adopted criterion of the minimum impact energy of KV_min_ 60 J. The impact test was carried out at room temperature. The Vickers hardness measurement was made with the indenter load of 10 kG (98.1 N)—HV10 using the QATM Qness 60 A+ hardness testing machine. The measurements were taken on the transverse microsection along three measurement lines determined next to the top and bottom face of the weld and in the central zone, respectively. The measurements were evaluated according to the adopted criterion of the maximum hardness in the area of the welded joint, HV10_max_ 450. The results of mechanical tests, i.e., static tensile test, presented in this paper were the average of three measurements, while the bend test included four measurements.

The macroscopic examination was performed on the transverse microsection including all zones of the joint: the weld, both HAZs and adjacent areas of the base material in the welded joint. The surface of the microsection was etched with the Barah reagent for approx. 20 s. To evaluate the macrostructure of the welded joint, 10× magnification was used in accordance with the adopted quality level B criterion.

The microscopic examinations were performed on transverse metallographic microsection etched with the Barah reagent. The duration of etching of the metallographic microsection prepared by grinding and polishing was approx. 10–15 s. The Barah reagent is commonly used to reveal the structure of duplex steels/cast steels [14]. The observation and recording of the images of the welded joint microstructure was carried out with the Keyence VHX 7000 digital microscope (DM). The volume percentage of austenite and ferrite in the test welded joint was determined by computer image analysis using the DM software.

## 3. Research Results and Analysis

### 3.1. Non-Destructive Tests

The visual tests confirmed that the test joint was produced properly, both on the top and bottom side of the weld. The evaluation of the joint meets the specified requirements in accordance with the adopted quality level B criterion. In turn, the penetrant tests revealed neither non-linear nor linear indications in both tested areas on the top and bottom side of the weld. The evaluation of the joint also confirmed the quality level B. Based on the radiographic test, the test joint was rated at the quality level B.

The positive results obtained during the non-destructive tests were the basis for further destructive testing to evaluate the properties of the test joint.

### 3.2. Macroscopic Examinations

The macroscopic image of the test X2CrNiMoN22-5-3 steel welded joint (Figure 2) shows that it has proper structure, i.e., correct penetration and regular fusion into both edges of the materials joined, proper arrangement of individual passes and beads, and small and mild weld face reinforcement. The particular weld fusion area of the first pass made by the hybrid method and remelted on both sides by the SAW method showed no deviations from the proper structure. The HAZ width was equal on both sides of the fusion line and amounted up to 2.5 mm at the bottom and top of the weld. In addition, no significant recorded welding imperfections were found on the cross-section of the joint in any of the assessed zones, i.e., in the base materials (BMs), HAZs and weld. The macroscopic examination of the X2CrNiMoN22-5-3 steel welded joint confirmed that the joint was made properly and with good quality.

### 3.3. Microscopic Examinations

Microstructure of the based material, i.e., X2CrNiMoN22-5-3 steel is presented in Figure 1. The test steel in the as-received condition had a two-phase microstructure consisting of banded dark-etched ferrite grains and bright-etched austenite grains (Figure 3). The austenite and ferrite grains were elongated in the rolling direction of plastic deformation. The microstructure of the test material was typical of this steel grade [4,11,14]. The volume percentage of ferrite in the base material was approx. 48%, and of austenite, approx. 52%.

At the fusion line (Figure 4), both on the face side and the fusion side of the welded joint, a section of the so-called incomplete fusion was visible where a smooth transition of austenite grains from the fusion area in the base material to the weld was observed. The weld in the analysed welded joint had a ferritic–austenitic structure typical of acicular/lamellar structures and Widmanstätten structures. No coarse-grained zone characteristic of this group of steels was observed close to the HAZ fusion line (Figure 5).

Generally, in the weld and the HAZ of the duplex steel, four austenite forms can be distinguished: allotriomorphic austenite precipitation on the grain boundary, side-plate Widmanstätten-like austenite, intragranular austenite phases and partially transformed austenite [5,7]. In the test joint, the following forms of austenite were observed both on the face side and the fusion side in the microstructure of this area: allotriomorphic austenite precipitation on the grain boundary (GBA), side-plate Widmanstätten-like austenite (WA) and intragranular austenite (IGA). The GBA and the WA were the predominant austenite phase forms in the analysed joint both on the face and fusion side (Figure 4). Similar observations of the microstructure of weld metal and HAZ were reported by other researchers [3,5,11,15]. The GBA nucleates and grows at the ferrite–ferrite boundary as energy-privileged sites. The precipitation of the GBA takes place at 1350 ÷ 800 °C [16]. Austenite grains grew along ferrite–ferrite boundaries forming thin layer of allotriomorphic austenite phase. The WA precipitates at 800 ÷ 650 °C, where diffusion is fast, nucleating directly on the GAB and/or partially transformed austenite. The WA formation requires a relatively small driving force and little undercooling [17]. The presence of the WA in the weld microstructure can be attributed to the presence of nitrogen in the chemical composition of the base material/filler metal [5,6]. In turn, the IGA precipitation takes place at 1000 ÷ 1100 °C in the chromium/molybdenum depleted zones and nickel/nitrogen-rich microareas at metastable ferrite [17].

The form of austenite nucleates heterogeneously on non-metallic inclusions, precipitates or dislocations and requires greater degree of undercooling [17,18]. On the face side of the weld, the ratio of the volume percentage of ferrite and austenite was around 55/45, while in the hybrid welding-affected area in the fusion zone it was 65/35 and in the remelting area—68/32. In the weld zone and HAZ (close to the fusion line), duplex steels solidify to a completely ferritic form, and the precipitation of austenite takes place in a solid form as a result of the following transformation: L → L + δ → δ + γ [16]. The formation of austenite phase is conditioned by the diffusion of stabilising austenite elements to this phase, while ferrite is enriched with elements stabilising this phase. The volume percentage of the austenite formed depends on the chemical composition of the steel, the cooling rate after heat treatment and/or welding process and the diffusion rate of diffusing elements. The increase in the cooling rate in the weld zone and HAZ can lead to a higher content of metastable ferrite due to inadequate time for transformation of ferrite phase to austenite phase [11,16,19]. Numerous austenite particles in the form of the WA (Figure 5) also indicates a high cooling rate in these areas of the weld. However, according to [5], the presence of the GBA and WA in the structure of the joint requires smaller driving force and less undercooling. Due to performance, it is not acceptable for industrial uses that the austenite volume percentage is less than 25% or the ferrite phase content is greater than 75%. The volume percentage of austenite in the weld meets the austenite content recommended for duplex steel joints of at least 25–30% [7].

### 3.4. Mechanical Testing

Figure 6 shows the Vickers hardness distribution on the cross-section of the welded joint. The measurements revealed that the BM hardness was between 259 and 270 H10, for HAZ it was 261–275 HV10, and in the weld—261–280 HV10. The obtained hardness on the cross-section of the welded joint was lower than the HV10_max_ 450 criterion. In general, the relatively small differences in hardness on the cross-section of the test joint (~260–280 HV10) were mainly related to the variation in ferrite-to-austenite ratio in the given area of the welded joint.

The highest hardness observed in the weld was due to the higher proportion of ferrite in this area compared to that in the BM or HAZ. Additionally, small ferrite grain in the microstructure of the welded joint has a significant effect on its hardness according to [19,20]. Hardness in the area of the welded joint which is comparable to that in the BM also indicates the absence of secondary phase precipitates affecting the increase in the precipitation hardening, resulting, among other things, in a significant increase in hardness. Additionally, the higher ferrite content in the weld compared to other areas of the analysed joint resulted in a relatively high tensile strength (TS) of approx. 790 ± 7 MPa. This value was higher by approx. 20% than the minimum TS required for this grade of material which is 660 MPa. The similar high TS values for duplex steel welded joints were observed in, but not limited to, Refs. [5,11,15]. Metastable ferrite in duplex steels is responsible for strength properties; hence, its higher content will result in an increase in these properties. In turn, according to [19,20], the fine ferrite grain in the area of the welded joint has a favourable effect on its strength. The effect of ferrite content in the test joint was also observed in the obtained value of impact energy (KV). The impact energy for samples with a notch cut in the weld axis (VWT 0/2) was 163 ± 4 J, and for samples with a notch in the HAZ (VHT 0/2) KV = 216 ± 26 J. The test material was characterised by high ductility, expressed by impact energy of 371 ± 8 J. Therefore, the impact energy results obtained for the weld and HAZ were approx. 44 and 58% of the roughness of the base material, respectively. The obtained results for impact energy were higher not only than the minimum requirements for this steel grade, but also than the data presented in, but not limited to, [4,5,17,21] for duplex steel welded joints. The toughness of a duplex steel welded joint depends on its microstructure, which is associated with the optimisation of the cooling rate, and the quantity and size of ferrite grain—fine grain increases the impact energy. The higher impact energy in the HAZ to was probably associated with the higher volume percentage of plastic austenite phase in this zone of the microstructure compared to the weld. The ductile austenite phase significantly prevents crack growth in contrast to the relatively fragile ferrite. Fine austenite boundaries in the HAZ also significantly prevent brittle crack growth [5,22]. According to [19], the high impact energy in the area of the welded joint of duplex steel can also be associated with the increase in the number of low angle grain boundaries (LAGBs). High proportion of LAGBs also has a positive influence on the corrosion resistance of duplex steel [23]. In turn, the tests in [24] show that not only fine grain, but also the grain boundary orientation has a characteristic influence on the impact energy of the welded joint of the duplex steel. The increase in percentage of austenite ∑3 coincidence site lattice increases the grain boundary impact energy of the welded joint [24]. On the other hand, the decrease in impact energy in the duplex steel welded joint area can be related to precipitates at the grain boundaries and/or inadequate transformation of ferrite phase into austenite phase in rapid air cooling from elevated temperatures after welding process [21]. The high value of TS and impact energy can also be significantly affected by the optimally selected welding filler metal [4,5]. The obtained values of impact energy in the welded joint zone were several times higher than criterion of KV_min_ = 60 J assumed for the examined material. The high plasticity and ductility of the joint produced were also evidenced by the results of the side bend test. The bend test of the welded joint showed the absence of macrocracks at the bending angle of 180° (Figure 7). This indicates the beneficial effect of the ferrite to austenite ratio in the volume of the welded joint on its plastic properties.

## 4. Summary

The thick-walled X2CrNiMoN22-5-3 duplex steel joint welded by the combined hybrid (laser + GMAW) and SAW method was subjected to physical metallurgy analysis. The joint was made as a double-sided joint with 2Y bevel in the central fusion zone using the hybrid (laser + GMAW) method, while the rest of the weld groove was filled by the SAW method. The investigations allowed for the following conclusions:The analysed welded joint has proper macro- and microscopic structure in all its areas, both in the fusion zone made by hybrid laser + GMAW method and in the filler welds made by the SAW method, which allowed the highest quality level B to be obtained for this joint.The analysed joint had high strength properties (TS ~ 790 ± 7 MPa) with good toughness in the range of ~163 ± 4 ÷ 216 ± 26 J and also a very good plasticity (bending angle of 180° with no cracks).The use of high-performance welding processes conducted under automated conditions using skilfully selected parameters allows to obtain a high-quality joint while ensuring high efficiency and beneficial reduction in production costs.The combination of hybrid welding processes (laser + GMAW) and SAW used in the paper has a large potential in terms of controlling parameters that affect the amount of heat input. The application of the developed welding technology in industrial conditions allows thick-walled duplex steel welded joints with the required quality level to be obtained.

## Figures and Tables

**Figure 1 materials-16-02909-f001:**
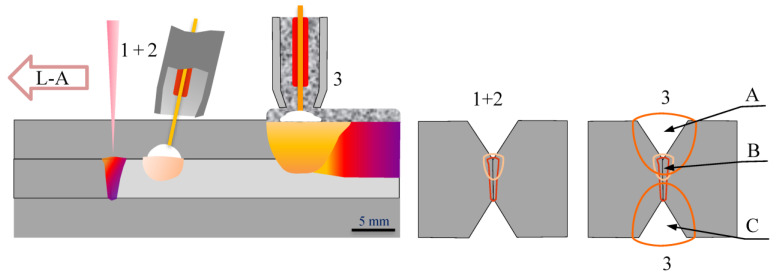
Diagram of the application of the hybrid + SAW process: (1 + 2) hybrid process (laser + GMAW), (3) SAW; A, B, C—microstructure investigation areas.

**Figure 2 materials-16-02909-f002:**
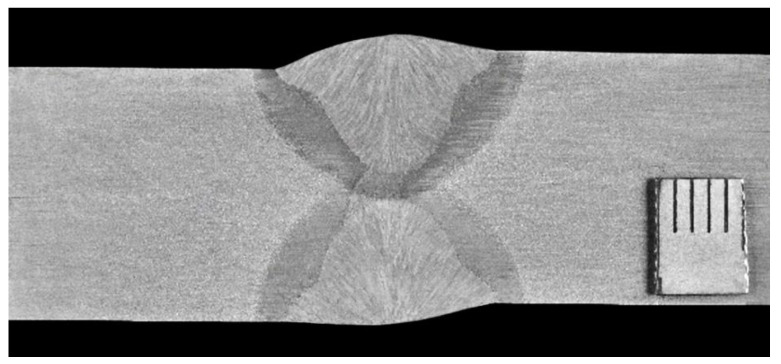
Macroscopic image of the cross-section of X2CrNiMoN22-5-3 steel welded joint.

**Figure 3 materials-16-02909-f003:**
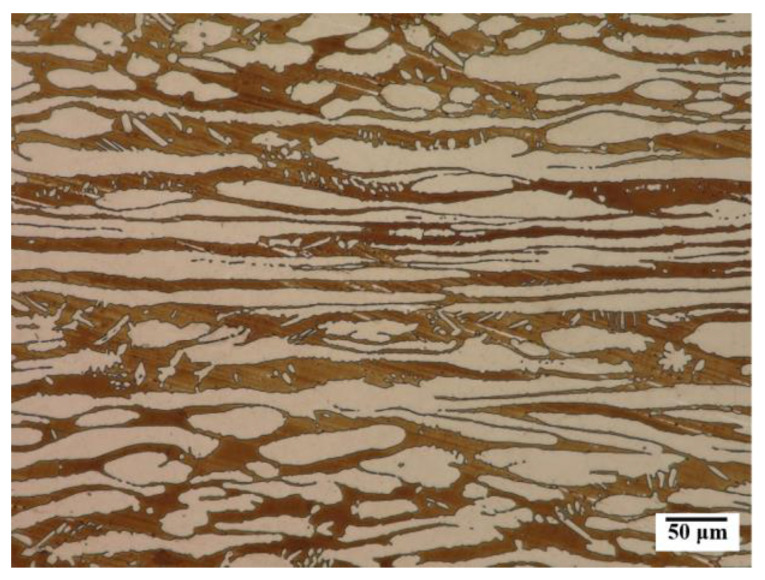
Microstructure of the base material.

**Figure 4 materials-16-02909-f004:**
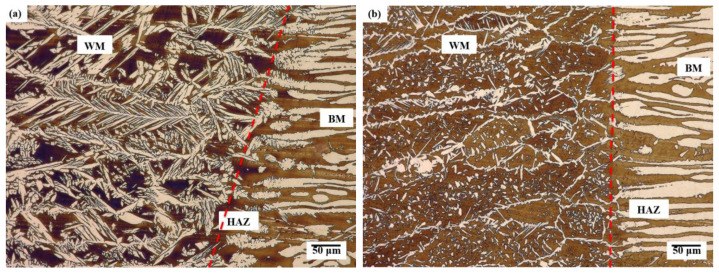
Microstructure in the fusion line on: (**a**) face side, (**b**) fusion side: WM—weld metal; HAZ—heat-affected zone; BM—base material, red dashed line—weld line.

**Figure 5 materials-16-02909-f005:**
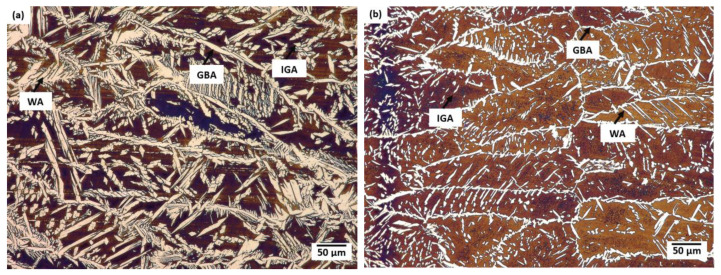
Microstructure of the weld in the area of: (**a**) face; (**b**) fusion, where: GBA—allotriomorphic austenite precipitation on grain boundary, WA—Widmansttaten-like austenite; IGA—intragranular austenite phase.

**Figure 6 materials-16-02909-f006:**
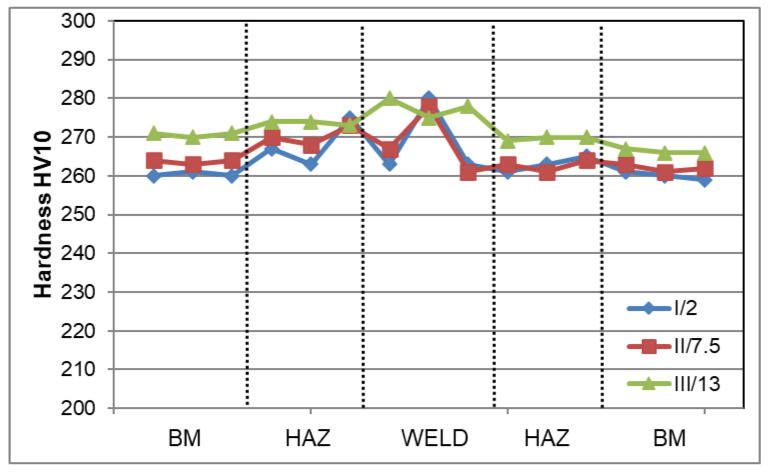
The results of the Vickers hardness measurement (HV10) on cross-section of the joint determined along lines from the top surface, where: I/2—measurement at a distance of up to 2; II/7.5—measurement line at a distance of 7.5 mm; III/13—measurement line at a distance of 13 mm.

**Figure 7 materials-16-02909-f007:**
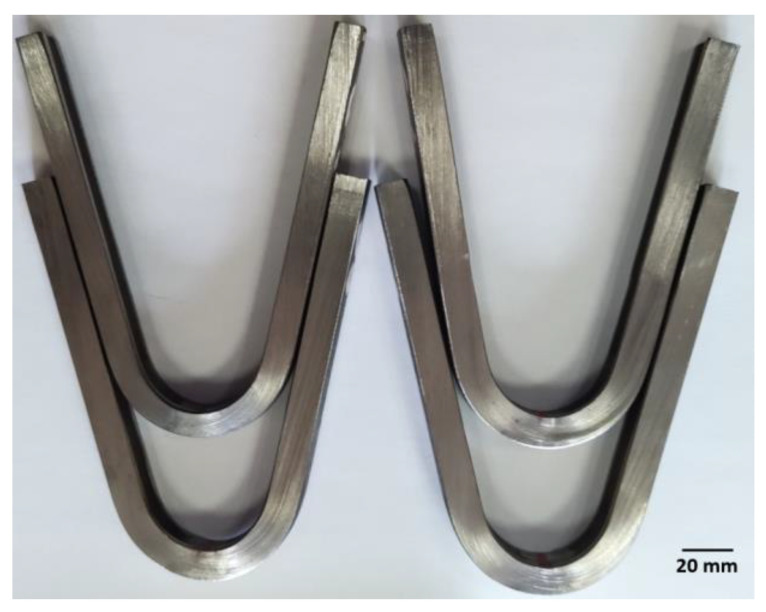
View of the samples after bend test.

**Table 1 materials-16-02909-t001:** The average chemical composition of X2CrNiMoN22-5-3 steel, wt. %.

Steel Grade	Average Chemical Composition, wt. %								
	C	Si	Cr	Ni	Mo	Mn	P	S	N
X2CrNiMoN22-5-3	0.011	0.53	22.8	7.21	3.14	1.29	0.022	0.002	0.11

**Table 2 materials-16-02909-t002:** Characteristics of the filler metal type of OK Autrod 2209.

Material Grade	Chemical Composition of Weld Metal, % wt							Properties of Weld Metal			
	C	Mn	Si	Cr	Mo	Ni	N	Tensile Strength N/mm^2^	Yield Strength N/mm^2^	El. %	KV_−20_ J
OK Autrod 2209	<0.08	1.50	0.50	22.5	3.2	8.5	0.15	765	600	28	85

where: El.—elongation; KV—impact energy determined at −20 °C.

## Data Availability

Not applicable.

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
