# Peer review of "Mechanical Properties and Microstructure of Austenite—Ferrite Duplex Stainless Steel Hybrid (Laser + GMAW) and SAW Welded Joint"

_materials, 2023, doi:10.3390/ma16072909_

Round 1

Reviewer 1 Report

Its an interesting research about DSS welding.

Some data is still missing (e.g. welding parameters) and a more depth evaluation must be made in some areas, but the manuscript is promising.

The literature recherche and results should be a little more separated, if possible.

I made a lots of comments and questions in manuscript with reiewers comments.

If corrected properly the manuscript can fulfill the publication criteria in Materials

Author Response

Dear Sir/Madam

The authors would like to thank for the valuable hints and comments, which contributed to the improvement in the quality of the study. All the comments/queries have been addressed; accordingly, the manuscript has been modified. The revised statements have been highlighted in ‘red’ for ease in identification Do hope the modified version will be suitable for publication in its present form. Awaiting to receive your valued decision,

Thanking You,

Sincerely Yours,

Grzegorz Golański

Reviewer 2 Report

The paper describes the microstructure and mechanical properties of duplex stainless steel hybrid (laser + MAG) and SAW welded joint. The information presented in the paper is valuable. Some minor revisions are recommended.

1.The authors should state what specification quality level B is based on.

2.The scale bar of Fig.4 and Fig.5 is not clear.

3.There are many symbols or abbreviations that should be corrected, e. g. % wt should be wt.%.

4.The same notation should be used, such as HV or HV 10.

5.Symbols or abbreviations should be clearly explained, such as VWT, VHT, KVmin, HV10min and 1350÷800℃, etc.

6.Page 6, line 190

Intragranular austenite (IGA) is missing the word “austenite”.

Author Response

(The authors gave the same response as above.)

Reviewer 3 Report

The presented manuscript “Microstructure and mechanical properties of duplex steel hybrid (laser + MAG) and SAW welded joint” needs major changes before taken into the considerations for publication:

·     Abstract is generic. Need to address a) need of this work, b) novelty, c) methodology suggested, d) results obtained (numerical).

·      Introduction section contains all the important aspects of the study. However, literature part was missing. Recent relevant work needs to be added.

·      Last para of Introduction does not address what is your present work is about, how you are addressing the problem, strategies etc...

·      Table 1 and Table 2 of section 2: Check the formatting and their appearance properly.

·      What is the basis of the selection of mentioned input conditions? Specify them in detail.

·      Figure 1: Add the scale bar

·      In conclusion, do not repeat the words that are used inside the manuscript. Do a complete revision and be precise to explain your key findings.

·      Compare the obtained results and justify the findings with proper technical reasons in results and discussion section.

·      Mention the limitations and further scope of improvement in last section.

Author Response

(The authors gave the same response as above.)

Round 2

Reviewer 3 Report

Accept in present form